# Simulated Microgravity Subtlety Changes Monoamine Function across the Rat Brain

**DOI:** 10.3390/ijms222111759

**Published:** 2021-10-29

**Authors:** Alexandra Gros, Léandre Lavenu, Jean-Luc Morel, Philippe De Deurwaerdère

**Affiliations:** 1CNRS, IMN, UMR 5293, University Bordeaux, F-33000 Bordeaux, France; alexandra.gros@u-bordeaux.fr (A.G.); leandre.lavenu16@gmail.com (L.L.); 2Centre National d’Etudes Spatiales, F-75001 Paris, France; 3CNRS, INCIA, UMR5287, University Bordeaux, F-33000 Bordeaux, France

**Keywords:** hindlimb unloading suspension, dopamine, noradrenaline, serotonin, neurochemistry, system biology, metabolite, correlative analysis, vestibular system

## Abstract

Microgravity, one of the conditions faced by astronauts during spaceflights, triggers brain adaptive responses that could have noxious consequences on behaviors. Although monoaminergic systems, which include noradrenaline (NA), dopamine (DA), and serotonin (5-HT), are widespread neuromodulatory systems involved in adaptive behaviors, the influence of microgravity on these systems is poorly documented. Using a model of simulated microgravity (SMG) during a short period in Long Evans male rats, we studied the distribution of monoamines in thirty brain regions belonging to vegetative, mood, motor, and cognitive networks. SMG modified NA and/or DA tissue contents along some brain regions belonging to the vestibular/motor systems (inferior olive, red nucleus, cerebellum, somatosensorily cortex, substantia nigra, and shell of the nucleus accumbens). DA and 5-HT contents were reduced in the prelimbic cortex, the only brain area exhibiting changes for 5-HT content. However, the number of correlations of one index of the 5-HT metabolism (ratio of metabolite and 5-HT) alone or in interaction with the DA metabolism was dramatically increased between brain regions. It is suggested that SMG, by mobilizing vestibular/motor systems, promotes in these systems early, restricted changes of NA and DA functions that are associated with a high reorganization of monoaminergic systems, notably 5-HT.

## 1. Introduction

Space programs are developing quickly, and the number of spaceflights is going to increase in the near future. Behind the technological exploit, spaceflights present several environmental challenges for living organisms. Radiations, isolation/confinement, and microgravity are some of the identified variables that may alter human physiology implying adaptive responses of the body, including brain homeostasis. In astronauts, spaceflights induce alteration of behaviors and cognitive functions, including the perception of orientation and movement, sensory integration, social interactions, learning and memory, and cognitive flexibility or mood [1,2,3]. Understanding the adaptive responses of the living organism to the spaceflight conditions represents a prerequisite for better preparation of astronauts before and after spaceflights.

Brain monoaminergic systems, including noradrenaline (NA), dopamine (DA), and serotonin (5-hydroxytryptamine, 5-HT), are neuromodulatory systems that are involved in the adaptation of living organisms to their environment [4,5,6]. These systems are innervating in a widespread manner and at various densities the whole brain from cell bodies located in the mesencephalon for DA (substantia nigra (SN) and ventral tegmental area (VTA)), caudal mesencephalon/pons for 5-HT (dorsal raphe (DR) and median raphe (MR) nuclei), and dorsal medulla for NA (locus coeruleus (LC)). Thus, these systems regulate all the functions mobilized by the organism during spaceflights (motor, vegetative, humoral, and cognitive functions). Acknowledging that monoaminergic systems are implicated in numerous neuropsychiatric and neurological diseases [7,8], it appears essential to determine how these systems respond to spaceflights conditions, in particular to microgravity. 

In astronauts, EEG or fMRI analysis showed modifications/adaptations of morphological and functional brain connectivity after long-duration spaceflights [9,10,11,12]. In rodents exposed to spaceflight (Cosmos 1129, Spacelab 3, LSM-STS-78 and Bion-M1 missions), few studies showed modifications of monoaminergic metabolism in several brain structures. It has been reported that a decrease of NA or 5-HT level occurs in some sub-regions of the hypothalamus [13,14]. Miller et al. reported an increase in the number of 5-HT_1_ receptors in the hippocampus and a decrease in the number of D_2_ receptors in the striatum [15]. In 1998, Blanc et al. analyzed, by high-pressure liquid chromatography (HPLC), the serotonergic system and observed modifications in the tryptophan, 5-hydroxytryptophan (5-HTP), and 5-HT levels in the prefrontal cortex, brainstem, hypothalamus, thalamus, striatum, cerebellum, and pineal [16]. More recently, RTqPCR analyses reported a decrease in the expression of D_1_ receptors in the hypothalamus and striatum, of the tyrosine hydroxylase expression in the SN, and of 5-HT_2A_ receptors in the hypothalamus [17]. These results indicated that monoaminergic systems seem to be impacted by spaceflight. However, the data obtained using this approach remains rare, conditioned by spaceflights with specific programs, and largely incomplete. 

Animal models have been developed in laboratories for years, in particular, the hindlimb unloading suspension model, which simulates microgravity on Earth [18]. This model reproduces in animals the principal effects of microgravity observed in human as atrophy in unloaded muscles, bone mineralization loss, and cardiovascular deconditioning [19]. It is associated with multiple molecular, cellular, metabolic changes of rat brain homeostasis and behavioral deficits [20]. Concerning the monoamines, RTqPCR analyses indicated modifications of the expression of monoamines receptors as an increase of the expression of D_1,_ D_2_, and 5-HT_2A_ receptors in the striatum and a decrease of the expression of 5-HT_2C_ receptors in the hippocampus after 30 days of suspension [21,22]. It was paralleled by a decrease of the expression of monoamine oxidase B in the midbrain and an increase of the expression of the catechol-O-methyl-transferase, the tyrosine hydroxylase, and the 5-HT transporter [21,22]. Using HPLC, a subtle region-dependent increase in 5-HT metabolism has been reported mainly in the hippocampus, hypothalamus, and prefrontal cortex with a decrease in DA metabolism in the prefrontal cortex and hippocampus [21,23,24,25]. Moreover, changes of tissue concentrations of monoamines were reported in the cortex and hippocampus along with memory deficits [20]. These data suggest region-dependent changes in monoamine levels induced by simulated microgravity (SMG), but the analysis of monoaminergic systems remains understudied. Indeed, the analysis is limited to a few structures/neurobiological networks, while some of the published neurochemical databases are questionable. 

Tissue assays of monoamines allow one to map through correlative analyses the possible reorganization of monoaminergic systems and make it possible to analyze a large number of brain structures [26,27,28,29]. Notably, brain structures associated with vestibular control, in addition to other systems classically studied, seem to be pertinent as the vestibular system, modified in astronauts [30], would display early changes from microgravity [31]. We postulate that SMG induces an early impact on monoamines in vestibular/motor systems with a reorganization of monoamine function across the brain. We explored, in this study, whether SMG over 7 days alters the distribution of monoamines within the brain of adult male Long Evans rats. Using HPLC coupled with electrochemical detection, we studied the concentration of the three monoamines and their main metabolites in 30 brain structures, including structures belonging to the vestibular system.

## 2. Results

### 2.1. Effect of Simulated Microgravity on Physiological Parameters

During the exposition to SMG, several physiological parameters were recorded. This daily follow-up of the animals suggested that SMG had moderate effects in rats (Figure 1). The weight of the animals at the start of the experiment was similar between CTL and SMG rats (206 ± 4.1 g and 205 ± 3.6 g, respectively). The ratio of the daily weight with respect to the initial weight over the seven days indicated that SMG rats had lower weight progression than control rats (two-way ANOVA, simulated microgravity effect, F(1, 22) = 27.54, *p* < 0.05; Figure 1A) and drank less water (two-way ANOVA, simulated microgravity effect, F(1, 22) = 26.67, *p* < 0.05; Figure 1B). Body temperature (37.3 °C in CTL vs. 37.3 °C in SMG, two-way ANOVA, F(1, 22) = 0.06, *p* = 0.8) and food consumption (22.5 g in CTL vs. 22.7 g in SMG, two-way ANOVA, F(1, 22) = 0.31, *p* = 0.58) were similar between groups. The measure of glycemia was performed at the beginning (J0) and at the end of the experiment (J7). Blood glucose was similar in both groups of rats (two-way ANOVA, simulated microgravity effect, F(1, 22) = 0.11, *p* = 0.74; Figure 1C). Collected blood at the end of the experiment permitted the concentration of corticosterone, a blood marker of stress, to be investigated. There was no difference between CTL and SMG groups (Student’s *t*-test, t_18_ = 0.47, *p* = 0.64; Figure 1D). 

### 2.2. Effect of Simulated Microgravity on Tissue Levels of Monoamines and Derivatives across the Brain

The effect of SMG on monoamine tissue content has been studied in 30 brain subregions. The regions were selected based on their assumed role in cognition, learning and memory, sensorimotor function and procedural learning, vegetative integration, posture, and equilibrium (vestibular). The sampled brain regions and their corresponding abbreviations are reported in Figure 2 and its legend. All the regions are associated with the main brain regions housing the cell bodies of ascending DA neurons (SN and VTA), NA neurons (LC), and 5-HT neurons (DR and MR) innervating the brain. 

#### 2.2.1. NA System

NA was measured in all sampled brain regions. Usually, low (striatal regions) to moderate concentrations were found in most brain regions. The highest concentrations were found in the HT, DR, and LC (Figure 3B right). No effect of SMG was observed in cortical regions (Figure 3A). In sub-cortical regions, SMG modified the content of NA in the IO, Cb, and SN (Mann–Whitney test, *p* < 0.05; Figure 3B). It is noteworthy that the changes corresponded either to a decrease (IO) or an increase (Cb and SN) in the NA tissue content. 

#### 2.2.2. DA System

The results concerning the quantitative analysis of DA and its metabolites DOPAC and HVA are reported in Figure 4A,B and the Appendix A
Figure A1, respectively. As expected [29], the concentration of DA was at the utmost levels in striatal sub-regions (Figure 4B right) and very low in several brain regions, including the cortices, DH, VH, T, and Cb (Figure 4A,B left). The lack of values in the DR and MR is due to the fact that the peak of DA was not clearly discernable due to another, unidentified compound that was co-eluted. The injection of the samples in another HPLC system could not resolve the issue. The DA content was similar in the SMG rats in most brain regions except in PL and RN (significant lower levels; Mann–Whitney test, *p* < 0.05), and S1A cortex (significant higher levels; Mann–Whitney test, *p* < 0.05). 

Tissue content of DOPAC was also quite similar between groups except in the PL and RSC (significant decrease in SMG rats; Mann–Whitney test, *p* < 0.05) and AMG (significant increase in SMG rats; Mann–Whitney test, *p* < 0.05; Appendix A
Figure A1A). The tissue content of HVA was also similar, although there was a significant decrease in PL, OB, and NAsh in the SMG group (Mann–Whitney test, *p* < 0.05; Appendix A
Figure A1B). The latter decrease is congruent with the non-significant trend toward a decrease observed for DA (Figure 4B middle) and DOPAC (Appendix A
Figure A1A) in the NAsh of the SMG group. 

Next, we analyzed the DOPAC/DA as an index of DA metabolism. As expected [28], the ratio DOPAC/DA in CTL was lower in the striatum (DM, VM, DL, and VL; Figure 4D) and higher in the other brain regions, with the highest value obtained in the EntC (Figure 4C). The DOPAC/DA ratio in the SMG group was similar to the CTL group except in IL (significant decrease in SMG rats; Mann–Whitney test, *p* < 0.05; Figure 4C) and RN (significant increase in SMG rats; Mann–Whitney test, *p* < 0.05; Figure 4D). 

#### 2.2.3. 5-HT System

5-HT was measured in all sampled brain regions with the highest concentrations found in VTA and SN (Figure 5B) and the lowest concentration found in the somatosensorily cortex (S1A, S1P and S1T; Figure 5A). The levels of 5-HT were decreased in the PL of SMG rats (Mann–Whitney test, *p* < 0.05; Figure 5A). The metabolite 5-HIAA was also measured in all sampled brain regions (Appendix A
Figure A2A). The quantities found in the S1T were closed to the limit of quantification. Similar to 5-HT, the levels of 5-HIAA were lower in the PL of SMG rats compared to CTL rats (Mann–Whitney test, *p* < 0.05). We analyzed the 5-HIAA/5-HT ratio as an index of 5-HT metabolism. This ratio was similar in all brain regions (Figure 5C,D) except in the DL, which was slightly higher in SMG rats (Mann–Whitney, *p* < 0.05; Figure 5D).

In this experiment, we measured tissue levels of 5-HTP (Appendix A
Figure A2B), the metabolic precursor of 5-HT. The signal in standard solutions was excellent, but we found that the endogenous levels were extremely low, requiring a high gain (10 nA/V). Since its elution time was in-between DOPAC and DA, we could measure its low levels in extrastriatal regions because DOPAC and DA were also studied at a higher gain. However, for striatal regions, the gain for DOPAC and DA was low (200 to 500 nA/V) because of their high quantities, impairing the determination of 5-HTP concentrations, which would be presumably low [32]. Indeed, whenever we got an electrochemical signal, we found that tissue concentrations of 5-HTP were very low in the somatosensorily cortex S1T (such as 5-HT and 5-HIAA) and PL, and highest in the MR (~150 pg/mg in CTL). In general, the concentrations ranged from 5 to 30 pg/mg of tissue in the CTL group. Its concentrations were lower in the IL and DH of SMG rats when compared to CTL rats (Mann–Whitney test, *p* < 0.05; Appendix A
Figure A2B).

### 2.3. Effect of Simulated Microgravity on the Qualitative Distribution of Monoamines and Derivatives across the Brain

The correlative approach of the neurochemical database allows for addressing qualitative changes of monoamines for a single monoaminergic system and between two compounds of homologous or heterologous systems in one or several brain regions. 

#### 2.3.1. Within a Single Monoaminergic Modality

The correlograms for NA, DA, and 5-HT elaborated for CTL and SMG groups are reported in Figure 6. As expected from previous works [28,33], the number of correlations across the brain for one monoamine was low. For NA, 33 correlations were found in CTL, five of them involving the LC with other brain regions (Figure 6A). The number of correlations was lower in the SMG group (23 correlations). The noticeable difference in the SMG group was the disappearance of correlations involving the LC and the enhancement of correlations involving the DR (1 in CTL vs. 4 in SMG). Moreover, the positive NA correlation between VTA and IL in the CTL group became negative in the SMG group (linear regression, r = 0.78, *p* < 0.05 in CTL vs. r = −0.88, *p* < 0.05 in SMG; inset Figure 6A). Similarly, 34 correlations were found for the DA content between regions (DR and MR were excluded due to our inability to get a correct electrochemical signal as explained above) in the CTL group and 25 in the SMG group (Figure 6B). Again, the five correlations involving the LC in the CTL group were not observed in the SMG group. Furthermore, the negative DA correlation between IO and RSC in the CTL group became positive in the SMG group (linear regression, r = −0.8, *p* < 0.05 in CTL vs. r = 0.62, *p* < 0.05 in SMG; insert Figure 6B). At variance with the catecholamines, 5-HT content tended to establish slightly more correlations between brain regions in SMG compared to CTL rats (24 correlations in CTL vs. 27 correlations in SMG; Figure 6C). There was a higher number of correlations involving the raphe nuclei, notably the DR in SMG compared to CTL group (one in CTL vs. four in SMG), mostly negative. The positive 5-HT correlation between SC and IL in CTL was lost in the SMG group (linear regression, r = 0.88, *p* < 0.05 in CTL vs. r = −0.04, *p* = 0.9 in SMG; insert Figure 6C).

The correlations of the DOPAC/DA and 5-HIAA/5-HT are reported in Figure 7A,B, respectively. For the DOPAC/DA ratio, 23 and 28 correlations were found in the CTL and SMG groups, respectively (Figure 7A). For example, the positive correlation between T and OFC in the CTL group became negative in the SMG group (linear regression, r = 0.86, *p* < 0.05 in CTL vs. r = −0.79, *p* < 0.05 in SMG; insert Figure 7A). We noticed a higher number of correlations of the ratio in the SMG group involving the HT (one in CTL vs. five in SMG) or the DH (zero in CTL vs. two in SMG). The most spectacular change between the maps of the two experimental groups was reported for the ratio 5-HIAA/5-HT (Figure 7B). While a few correlations were reported in CTL, the number of correlations highly increased in the SMG group (20 in CTL vs. 55 in SMG). It notably involved the ratios of the EntC (10 correlations), DR (9 correlations), PL (9 correlations), and VL (9 correlations). For example, the 5-HIAA/5-HT correlation became positive in SMG rats (linear regression, r = 0.17, *p* > 0.05 in CTL vs. r = 0.84, *p* < 0.05 in SMG; insert Figure 7B). 

#### 2.3.2. Between Indexes across the Brain

We focused on the correlations between compounds in a single brain region (Figure 8) and only one full display concerning the correlations established between the ratios DOPAC/DA and 5-HIAA/5-HT across the brain (Figure 9) to illustrate the changes of relationships between the compounds/systems across the brain induced by the SMG. 

We first addressed the relationships between related compounds in the same brain region (Figure 8). In the CTL group, the DA content correlated with the content of its metabolite DOPAC in half of the sampled regions (Figure 8A), while it correlated approximately in a quarter of the regions with the content of its other metabolite HVA (Figure 8B). The DOPAC content also correlated with its direct product HVA in 13 brain regions (Figure 8C). The relationship parent compound/metabolite was slightly different in SMG condition (fewer numbers of correlations for DA and DOPAC: 15 correlations in CTL vs. 10 correlations in SMG, Figure 8A; higher numbers of correlations for DA and HVA: 7 correlations in CTL vs. 10 correlations in SMG, Figure 8B; and for DOPAC and HVA: 13 correlations in CTL vs. 15 correlations in SMG, Figure 8C). Several correlations persisted between the two conditions, notably in the LC and the VTA, whereas the main distinctions in the pattern of correlations were found at the level of the OFC (loss in SMG), PL (increase in SMG) IL, RN, SC (increase or appearance in SMG), and DL.

The changes in catecholaminergic function reported above led us to study the correlations between NA and DA (Figure 8D), DOPAC (Figure 8E), or HVA (Figure 8F). Varying from previous analyses from the lab [26,28], the correlations between NA and DA compounds were largely present in the CTL group, and their number was increased in the SMG group (9 correlations in CTL vs. 13 correlations in SMG; Figure 8D). PL, Cg, S1P, S1T, RN, SC, IO, AMG, DM, DL, VL, SN, and VTA were the structures exhibiting the changes in correlations. Whereas the number of correlations between NA and HVA did not change between the CTL and SMG groups (12 in both groups) with a few modifications of the pattern (Figure 8E), it was doubled in the SMG group between NA and DOPAC (8 correlations in CTL vs. 16 correlations in SMG; Figure 8F). For the latter, the correlations notably appeared in regions belonging to the vestibular system (IO), primary sensorimotor cortex (S1P), and prefrontal cortex (PL and IL). The content of 5-HT correlated with the content of its metabolite 5-HIAA in 17 regions in CTL and 18 in the SMG group (Figure 8G). The pattern of correlations was slightly modified in all systems, with more pronounced changes found in the cortical regions (OFC, IL, Cg, and EntC).

Second, we addressed the correlations between the ratios of DOPAC/DA and 5-HIAA/5-HT, the content of NA and DA, DA and 5-HT, and 5-HT and NA across the brain. The number of correlations between the ratios was lower in CTL compared to the SMG group (47 correlations, including 19 negatives, in CTL vs. 95 correlations, including 24 negatives in the SMG group; Figure 9). The gain in the number of correlations in the SMG group concerned several functional territories and notably the cortex and the vestibular system. We can highlight that the ratio pf DOPAC/DA from the EntC, RSC, NAco, T, or OFC, to cite a few, established several correlations with the ratio of 5-HIAA/5-HT across the brain in the SMG group. The ratio of 5-HIAA/5-HT from the PL, RSC, striatum, or HT also correlated more with the brain DOPAC/DA in the SMG group compared to CTL (Figure 9). The correlations between neurotransmitters in the two experimental groups are illustrated in the Appendix A
Figure A3. For the three correlative analyses concerning NA and DA, DA and 5-HT, and 5-HT and NA, we found an approximately similar number of significant correlations between the CTL and SMG groups (always lower than 10% of the total correlations performed). Yet, the number of correlations was always lower in the SMG group, with a slightly higher proportion of negative correlations (reported in the Appendix A
Figure A3). For NA vs. DA, the pattern of correlation was different, and we noticed in the SMG group fewer correlations in the LC and more correlations from the NA in the DR/MR with DA, all of which were negative (Appendix A
Figure A3A). The 5-HT content from cortical regions established fewer correlations with brain DA content in the SMG group, with the exception of the S1A somatosensorily cortex. DA content in the basal ganglia (striatum including NAsh and NAco), DH, or AMG had fewer correlations with brain 5-HT content, with the exception of the vestibular system (Appendix A
Figure A3B). Finally, the pattern of correlation between NA and 5-HT in the brain was also modified in the SMG group. NA content in the cortex of SMG rats correlated less with brain 5-HT content, whereas 5-HT content in the basal ganglia correlated less with NA content in the somatosensorily cortices and vestibular system. The NA content of the DR correlated much more with brain 5-HT in the SMG condition (Appendix A
Figure A3C). When looking at the correlations of monoamines in single brain regions (diagonals of the Appendix A
Figure A3), we noticed that the content of all monoamines correlated in the OFC, IL, and AMG and never correlated in the HT whatever the condition (CTL or SMG). 

## 3. Discussion

In our study, we have investigated the impact of short-duration exposure to simulated microgravity on monoaminergic neurochemistry in multiple brain regions of adult rats. While the modifications were very few, they were highly specific. We found that NA tissue content was modified along brain regions of the vestibular system and motor system; the markers for the DA system were modified in RN, PL, NAsh, AMG; and 5-HT tissue content was specifically decreased in the PL. These small quantitative changes were paralleled by substantial modifications of the pattern of correlations of tissue content of monoamines across the brain, with the highest changes concerning the 5-HT system. It is concluded that SMG over a short period (7 days) triggers specific, restricted changes of catecholamines and 5-HT and already remodels the organization of monoaminergic systems in the brain.

We confirmed the slight physiological modifications previously observed under SMG in laboratory animals: (1) the stop of the natural increase body weight even for a small period of 7 days, also observed during spaceflight [34,35,36,37,38], (2) the decrease of water intake only during the first 1–3 days of suspension [35,39,40], and (3) the absence of alteration of glycemia [41], food intake [34], and body temperature [34]. The impact of SMG on plasma corticosterone level is under debate and probably reveals hypersensitivity to stressors related to suspension. Some studies report an increase in corticosterone after 7 days of suspension [23,42], while we and others did not detect a difference [43]. Differences between all of these studies could be due to other experimental parameters, such as the rat strains or environmental conditions of the animal facilities. In addition, the poor change of monoamines content we report in brain regions involved in stress responses, such as the hippocampus, the hypothalamus, the amygdala, or the anterior cingulate cortex, was also suggestive of the non-stressful conditions of the procedure of suspension, in line with the conclusions offered by previous authors [13,14,15,24]. 

We found that the NA content was significantly changed in some brain regions of the vestibular system (Cb, IO), as well as the SN, suggesting that SMG mobilizes the vestibular system and promotes changes in NA neuromodulation. Such an action would concern local changes rather than a general response of the activity of NA neurons of the LC because (1) the changes were opposite in Cb and IO, and (2) one would have expected multiple changes across the brain in the case of an unspecific response. In the same vein, the DA tissue content was also reduced in RN (increased of the ratio DOPAC/DA), whereas it was increased in the part of the somatosensorily involved in the representation of the anterior limb (S1A). In keeping in mind that the 5-HT system was not modified in the vestibular and motor systems, it appears that the net, restricted changes of monoamines in those motor/vestibular networks concerned catecholaminergic function. That catecholamine contents are sensitive to the SMG in those regions is consistent with data reporting changes of the vestibular system in astronauts after short and long-duration spaceflight [30,31].

The extent to which the DA changes are related to DA neurons terminals remains uncertain because DA is the metabolic precursor of NA, and the activity of the NA neurons can mediate DA effects [44,45,46,47]. One advantage of performing correlative analyses is to address in a single brain region the relationship between the compounds of homologous systems, for instance, DOPAC vs. DA content, and presumably heterologous compounds, such as NA and DOPAC. In our growing experience of this type of analysis [26,28,29,33], although performed in strains and species other than Long Evans rats, the relationship between NA and the DA metabolites in several brain regions is usually lower than that reported in the present experiment. First, we found some correlations between NA and the DA metabolites that could be related to our “control” conditions (see below), and second, the number of these “heterologous” correlations increased in the SMG group. In addition, the correlations between DA and NA or DOPAC and NA were enhanced in SMG conditions in some regions of the vestibular/motor system (SC, RN, IO). A similar situation has been noticed at the level of the PL or the AMG where the decrease of DA, DOPAC, and HVA content or the increase of DOPAC content, respectively, was associated with a higher appearance of correlations between these compounds and NA. Consequently, we cannot affirm that the changes of DA, DOPAC, or HVA tissue content in the abovementioned brain regions correspond to DA neurons working together with NA neurons (to account for the enhancement of correlations in SMG). It could suggest that NA neurons are handling all the aspects of catecholaminergic neurochemistry that we report alone, or it could reflect the combination of the biochemistry of both DA and NA neurons.

Conversely, DA content from the NAsh to the DL striatum corresponds to the activity of DA neurons, as the DA content and the DA fibers are tremendously high compared to the markers of the NA system [28]. We found a decrease in HVA content specifically in the NAsh, which was congruent with the non-significant trend toward a decrease of DA, DOPAC, and the ratio of DOPAC/DA. It is noteworthy that DA indices, including DOPAC/DA, were poorly modified in the striatum between CTL and SMG, either quantitatively or qualitatively, as suggested by the correlative analysis. According to the hypothesis of ascending loops connecting the NAsh to the DL striatum proposed by Haber et al. (2000), the decrease in DA tone in the NAsh would limit this loop process [48] and procedural learning. Meanwhile, a decrease in DA function in the NAsh would suggest a loss of motivation in the SMG group. It would be interesting to test the different aspects of motivation in rats in several tasks as in appetitive instrumental learning [49].

It was surprising that the tissue levels of 5-HT or its metabolite 5-HIAA were almost not significantly modified across the brain, with the noticeable exception of the PL cortex (lower levels for both compounds in the SMG condition). In line, mice exposed to a 1-month spaceflight had very few changes of key enzymes/transporters/receptors expression of the 5-HT system in the brain, at variance with the DA system [17]. The other advantage of associating a correlative analysis of the neurochemical database is to propose that the quantitative aspects of the 5-HT system are maintained despite the changes in connectivity. Indeed, at variance with the catecholaminergic systems, we report a net increase in the number of correlations for the ratio 5-HIAA/5-HT in the SMG condition. It involved the DR, as well as other structures, such as the EntC, PL, and VL, which displayed correlative links with numerous other brain regions. The higher connectivity of the DR in SMG rats was not only noticed for 5-HT, 5-HIAA, and their ratio, but also for NA or HVA to a lower extent. The trend towards a decrease in the correlative map of DA markers associated with an increase of the correlations for the 5-HT markers has already been encountered after the injection of the two preferential 5-HT_2C_ receptor agonists, lorcaserine and WAY-163909 [27,50,51], two compounds that can lower the VTA DA neuron firing rate [51,52]. 

The finding that SMG can have an effect on monoamine tissue contents has already been reported in the literature [20,23,24,25]. Notably, 15 days of hindlimb unloading suspension of rats had no consequence on NA, DA, and 5-HT tissue content in the hippocampus, striatum, hypothalamus, nucleus accumbens, and prefrontal cortex, but increased the 5-HIAA/5-HT ratios in the hippocampus and the hypothalamus and decreased the DOPAC level in the prefrontal cortex [24]. In our study, we confirm that the consequence of the SMG on brain monoamines are specific to the considered monoamine and are restricted to some brain regions. Furthering the comparison between these studies and our work is conditioned by three main factors: (1) the duration of hindlimb unloading suspension, (2) the size of sampled tissue, and (3) the validity of the neurochemical measurement. First, the delay of hindlimb unloading suspension was short in our experiment, which allowed us to evaluate the early reorganization of monoaminergic systems. The changes in the correlation pattern of 5-HT markers across the brain might be the premise of quantitative changes occurring later. In that way, it has been reported that longer exposure to SMG induced stronger modifications of 5-HT, DA, and NA tissue content in the hippocampus and/or cortex [20,21,25]. It would be interesting to evaluate the effect of long-term exposure to simulated microgravity in a large number of brain structures. Second, our sampling method allows us to distinguish subregions that could respond differently. Notably, the PL, IL, and Cg encompass the “prefrontal cortex”, but we report that these structures respond differently to the SMG as in other situations, such as impulse control or decision-making-related responses [7,53,54]. The specific decrease in DA and 5-HT content we report in PL might be unseen when encompassing the other parts, acknowledging that some authors reported a decrease in DOPAC content as we did [20,24]. Conversely, our sampling method could have masked specific subregional effects, such as in the HT. Indeed, spaceflight has been shown to alter NA content in discrete regions of the rat hypothalamus, an effect that is not reported when taking out the whole structure [14]. Third, the comparison between studies cannot be done when the absolute values reported in previous studies are outside the range of the concentrations that are normally reported in rodents and mammals. In some studies [20,23,25], the tissue levels for some monoamines, particularly DA in the hippocampus or the frontal cortex, are aberrantly high intrinsically [20,23], or when compared to 5-HT and NA contents [20,25], which should be higher. It could come from a bad sampling of tissue (contamination of the striatum in the cortex, for example), an error in the determination of the analytical peak (confounding eluent/ions), and/or some calculation errors from the calibration curves. In any case, the monoamine tissue contents were not properly reported in these studies, which impairs any comparison from our side. As regards 5-HTP, it has been previously reported that spaceflight was lowering its concentrations in several rat brain regions [16]. However, the basal concentrations of 5-HTP found in the latter study were similar to the values of 5-HT and its metabolite, at odds with our and previous values of the literature, which report very low to undetectable values of 5-HTP [32,55]. 

While the main changes we report concerned sub-components of the vestibular motor, and cognitive systems, the modifications in brain regions involved in learning and memory, procedural learning, and stress responses were very few (RSC, EntC, hippocampus, amygdala, striatum). Keeping in mind that the vestibular system has a substantial impact on cognitive or executive function [56], it is possible that the SMG exposure in our study was not long enough to quantitatively detect these changes in these structures. The distinct correlation profile in SMG rats, notably as regards the 5-HT system and the 5-HIAA/5-HT vs. DOPAC/DA ratios, suggests that the remodeling of the central 5-HT transmission involves the EntC, the VL, and, to a lower extent, the HT and the RSC. Moreover, our control conditions could already alter the activity of neurobiological networks involved in cognition, mood, and memory. Indeed, the notion of controls in the study of the microgravity effect deserves caution, and we carefully selected the independent variable “isolation” as one confounding factor. Meanwhile, it is likely that the isolation, although operating for a small period, could alter the responsiveness of monoaminergic systems [57,58]). It could already impact the relationship between the NA and DA systems we exemplified above, as well as the levels of correlations of the ratio 5-HIAA/5-HT that was found to be higher in the control groups of other experiments [26,33]. 

Even if the analysis concerned thirty regions or sub-regions of the brain, it is still limited to have a full overview of the dynamic of monoaminergic systems under SMG. Moreover, we did not explore their consequences on cerebral plasticity or cognitive and motor behaviors. Yet, it is sufficient to show that the changes consequent to SMG are (1) specific to each monoaminergic system, (2) restricted to a few brain regions, and (3) have consequences on the global mapping of monoamines interactions across the brain. It is suggested that monoamines adjust their modulatory role according to the demand inherent to the SMG (here, vestibular/motor and operating PL cortex) and tune the consequences on other networks. However, this study questions the putative role of monoaminergic systems to better understand the effect of longer exposure to microgravity on brain functions and also the interindividual differences in the physiological response to microgravity. The allostatic status conferred by these adaptations might not be a problem for astronauts unless their return to Earth is devoid of a specific action. Additional studies are warranted to determine the impact of the other factors characterizing spaceflight, including isolation and irradiation, on brain physiology, and also to test the effects of proposed countermeasures to limit the physiological alterations due to space exploration. 

## 4. Materials and Methods

### 4.1. Animals

Adult male Long Evans rats (Janvier Labs, 7 weeks on arrival, *n* = 24 rats equally distributed in two groups: 12 in an isolated cage as controls (CTL) and 12 in simulated microgravity group (SMG)) were used in the study. The animals were initially housed in groups, 2 per cage, according to the calculation of the size of the cage in regard to their weight (model GR900, 395 × 346 × 213 mm, surface: 904 cm^2^, Techniplast) and were maintained under standard conditions with 12-h light/dark cycles (lights on at 7:30 AM) at room temperature (21 ± 2 °C, 60 ± 5% humidity). They were fed a standard pellet diet (A04 product, SAFE) and tap water ad libitum during the study. The rats were handled once daily in order to avoid the stress and to familiarize them with the male and female experimenters. All the procedures used were in accordance with the European Communities Council Directive (2010/63/EU Council Directive Decree) regarding the care and use of laboratory animals. The procedures were approved by the Ethical Committee (CEEA-050, project #28854) of Centre National de la Recherche Scientifique (CNRS) and the University of Bordeaux.

### 4.2. Procedure for Simulated Microgravity

After handling for 10 days, for acclimation to the vivarium conditions (UMR 5293 Institute, University of Bordeaux; agreement A32-063-940), the rats were divided into two groups: one control group (CTL, *n* = 12) and one group submitted to simulated microgravity (SMG, *n* = 12) for 7 days. To simulate microgravity, we used the hindlimb unloading rat model, modified from the Morey-Holton method [18,19,59]. Briefly, the tails of the rats were cleaned and protected with medical adhesive tape. A metal ring with a flexible cord was attached to the rat’s tails with tape. The cord was clipped on a pulley attached to a mobile metal bar inserted in the roofs of the cages. Rats were able to move freely in the cage with their forelimbs in a 360° arc. The hindlimbs were kept above the cage floor. The head-down tilt position was near 40–45°. After an acclimatization period in individual cages over 3 days, the hindlimb-suspended rats were maintained in this position for 7 days. The animals were monitored daily to ensure access to water and food and that their hindlimbs were not touching the grid. The control rats were kept isolated in standard cages with an identical grid to suspended cages to ensure similar sensory inputs between conditions. All animals were kept in the same experimental room. 

### 4.3. Physiological Measurements

The weight and body temperature of the rats and their food/water consumption was taken on a daily basis. The glycemia was measured in blood with the Accu-check Performa device (Roche), just before hindlimb unloading suspension and just after the end of the suspension procedure 7 days later. Blood was collected by puncturing the end of the tail of each animal. The corticosterone concentrations were measured by ELISA (K014-Arbor Assay LLC) in the supernatant of the blood collected after rat decapitations and after centrifugation at 10,000 rpm for 5 min at 4 °C. The supernatant was stored at −20 °C until use.

### 4.4. Tissue Collection of Brain Regions

After 7 days of suspension (SMG) or in control (CTL) cages, rats were immediately brought to the sacrifice room. They were killed by decapitation using a guillotine and without prior anesthesia; anesthetics alter the activity of monoaminergic systems. The biochemical processes we studied are fast, so anesthetics thereby represent a strong confounding factor in interpreting the data [33]. The brain was rapidly removed and immediately placed in cold isopentane (−50 °C for 3 min). The brains were stored at −80 °C until use.

A cryostat (Leica CM3000, Leica Biosystems, France) was used to collect brain regions of interest. The cryostat chamber was maintained at −24 °C. Brain regions were taken with the help of a rat brain atlas [50] and a magnifying glass. Homemade stainless steel cannulae of 500 or 800 µm inner diameters were used to collect the brain regions, which were deposited in labeled, pre-weighed small Eppendorf tubes (0.6 mL volume). Of note, for a brain, the left and right sides of a brain region were pooled except for the MR and DR, which consisted of one punch encompassing the two sides. Once collected, the tubes containing the tissue were stored at −80 °C.

Figure 2 depicts the sampled brain regions. Except for the olfactory bulbs (OB), 29 distinct brain regions were taken out with the cryostat, including cortical regions (orbitofrontal cortex (OFC), prelimbic cortex (PL), infralimbic cortex (IL), anterior cingulate cortex (Cg), retrosplenial cortex (RSC), entorhinal cortex (EntC), and three distinct somatosensoriel cortices for the forelimb (S1A), the hindlimb (S1P) or trunk (S1T)), subcortical regions involved in procedural learning (the nucleus accumbens shell and core (NAsh and NAco), four parts of the striatum including dorsomedial striatum (DM), dorsolateral striatum (DL), ventromedial striatum (VM), ventrolateral striatum (VL)), subcortical regions involved in memory and emotions (the nuclei of the amygdala (AMG), the dorsal and ventral parts of the hippocampus (DH and VH)), the thalamus (T) corresponding to the ventrolateral parts, the hypothalamus (HT), the brain regions containing cells bodies of monoaminergic systems (substantia nigra (SN), the ventral tegmental area (VTA), the dorsal and median raphe nuclei (DR and MR), and the locus coeruleus (LC)), and brain regions being somehow involved in automatic posture and equilibrium (the cerebellum (Cb) corresponding to part of the vermis, the superior colliculi (SC), the red nucleus (RN), and the inferior olive (IO)). The traces left after having collected brain regions were systematically pictured using the camera of a smartphone. 

### 4.5. Tissue Processing and Neurochemical Analysis

The tubes containing the brain tissue of one region were taken out from −80° and placed on ice. In order to measure the weight of the tissue, the tubes were carefully and quickly wiped and weighed on the same precision balance [26]. Immediately after the weighing, 100 µL of 0.1 *n* HClO_4_ (4 °C) were added into the tube and tissues were homogenized using sonication. The tubes were centrifuged at 13,000 rpm for 30 min at 4 °C (Eppendorf 5424R). Depending on the studied structure/chromatographic system, aliquots of 10 or 20 µL of the supernatants were directly injected into the HPLC system coupled with electrochemical detection (HPLC-ECD). For olfactory bulbs, due to the large weight of the tissue (21 mg instead of 1–4 mg for the other brain regions), 200 µL of HClO_4_ were initially added into the tubes, and the supernatant was diluted (1/2) in the mobile phase of the corresponding HPLC system. A total of 10 µL of the dilution was injected into the HPLC system.

### 4.6. Chromatographic Analysis

The HPLC-EDC systems were used to measure the tissue concentrations of NA, DA, and 5-HT, as well as two 5-HT-associated compounds (5-HTP, the precursor, and 5-HIAA, the metabolite) and two metabolites of DA (DOPAC and HVA). The HPLC system that was mostly used was composed as follows: a mobile phase (70 mM NaH_2_PO_4_, 0.1 mM disodium EDTA, 100 µL/L triethylamine, 2-Octane-sulfonic acid approximately corresponding to 130 mg/L of mobile phase (see below) in deionized water (18 MΩ.cm^−2^) containing 7% methanol), a manual injector equipped with a 20 µL loop (Rheodyne 7725i, C.I.L.-Cluzeau, Sainte-Foy-La-Grande, France), a HPLC column Equisil ODS (C18) (150 × 4.6 mm, 5 µm; C.I.L.-Cluzeau), and a HPLC pump (LC20-AD, Shimadzu, France). The temperature of the column was maintained at 40 °C using an oven. Before its installation into the system, the mobile phase was filtered using a 0.22 mm Millipore filter. Once installed, the mobile phase was delivered at a 1.400 mL/min constant flow rate. The pH of the mobile phase, using orthophosphoric acid (85%) and the concentration of 2-Octane-sulfonic acid, was adjusted to get the best separation between electrochemical reactive eluents. We considered a good separation of the abovementioned eluents plus other molecules, such as adrenaline, 3-O-methyl-DOPA, 3-methoxytyramine (3-MT) (a metabolite of DA), three metabolites of NA (vanylmandelic acid, dihydroxyphenylethylene glycol, and 3-Methoxy-4-hydroxyphenylglycol), octopamine, tyramine, tyrosine, and tryptophan. In most cases, those molecules did not produce clear electrochemical signals either because the potential set at the electrodes was not sufficient and/or their endogenous concentration was too low. It was the situation of the peak of 3-MT that was overlapping with the peak of 5-HT. In most brain regions, the signal produced by 3-MT is negligible compared to 5-HT, but it is not the case in striatal regions. Thus, we had to analyze the striatal samples (including the nucleus accumbens) on another HPLC system. Again, the peaks of 3-MT and 5-HT were almost co-eluted, but we could analyze them separately with distinct potentials (see below). 

The monoamines were eluted from the column at different retention times (approximately: NA, 2.6 min; DOPAC, 4.3 min; 5-HTP, 5.7 min; DA, 6.6 min; 5-HIAA, 8 min; HVA, 10.9 min, and 5-HT, 17.2 min), which were then entered the coulometric detection cell (Cell 5011, ESA, Paris, France) equipped with two electrodes. For the most used HPLC system (for extrastriatal regions), we fixed the potential of the two electrodes at +350 and −270 mV on the coulometric detector (Coulochem II, ESA, Paris, France), but we only used the signals obtained in oxidation. The electrochemical signals were acquired at different gains depending on the compound and the studied brain region (from 10 to 200 nA/V) during the chromatogram using a timeline method recorded in the coulometric detector. The detector was connected to an interface that conveyed the signals to the computer (Ulyss, Azur system, Toulouse, France). The system used for striatal samples was similar, except two electrochemical cells (implying two coulochem detectors) were placed in series. The first potential of oxidation was 200 mV on the first cell (sufficient to allow for the oxidation of 5-HT but not 3-MT and not optimal for HVA). Then, after the passage through the electrode of reduction (−150 mV), the third electrode was set at +400 mV, mainly to acquire the data for HVA. The procedure has been previously published [60].

Calibration curves were performed using a range of concentrations of eluents compatible with the expected quantities. Standard solutions containing all the compounds of interest at known concentrations were systematically injected each day before and after a series of samples.

### 4.7. Statistical Data Analysis

All procedures from the brain dissection under the cryostat to the measurement of monoamine contents by HPLC have been performed blind of the treatment and randomly. The results for individual compounds are expressed in pg/mg of tissue. The ratios of DOPAC/DA and 5-HIAA/5-HT, considered as an indirect index of the turnover, were also calculated. The mean ± SEM of values in the two groups are reported in the graphs. Aberrant data were removed based on the ROUT method [61] developed by GraphPad to identify outliers from nonlinear regression. 

In several cases, the distribution of the data was not normally distributed using the Shapiro–Wilk test, and we decided to use the non-parametric Mann–Whitney test for comparing the tissue levels of monoamines or metabolites between the two groups (CTL vs. SMG). For the same reason, we used the Spearman rank-order test for studying the correlations. For the quantitative analysis studied with the Mann–Whitney test, *p*-values were adjusted using the False Discovery Rate (FDR) controlling procedures. Significance was considered at the 5% level. All statistical data analysis was performed using GraphPad Prism, version 9.2.0 (GraphPad Software, LLC).

## 5. Conclusions

We report the physiological consequences of 7-day SMG on the distribution of monoamines in functional territories of the rat brain. SMG induces a few quantitative changes that are regionally restricted and that mainly concerned catecholaminergic function. Nonetheless, the qualitative changes assessed with correlative analyses clearly show remodeling of the 5-HT system alone or in interaction with catecholaminergic systems, notably DA. The reported profile induced by SMG might not be pathogenic per se but could likely confer transient vulnerability to develop neuropsychiatric diseases, notably mood disorders. Our data strongly support specific programs of rehabilitation for astronauts for their return to Earth.

## Figures and Tables

**Figure 1 ijms-22-11759-f001:**
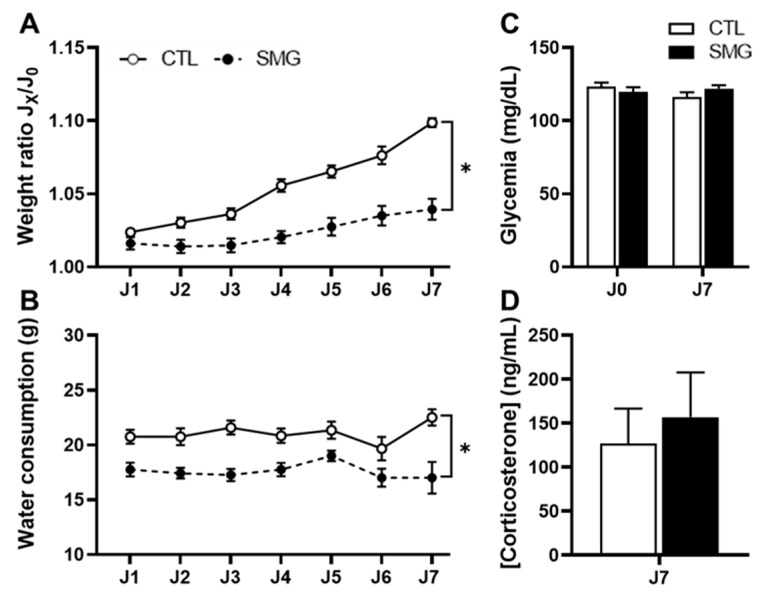
Physiological parameters of the rats during the experiment. (**A**). Ratio of the daily weight with respect to the initial weight over the 7 days of SMG. (**B**). Water consumption over the 7 days of SMG. (**C**). Glycemia measured at the beginning (J0) and at the end (J7) of the SMG. (**D**). Corticosterone concentrations measured in blood collected at J7. All data are presented as mean ± sem (*n* = 12 rats/group). * *p* < 0.05 (corresponding to two-way ANOVA followed by post-hoc test results).

**Figure 2 ijms-22-11759-f002:**
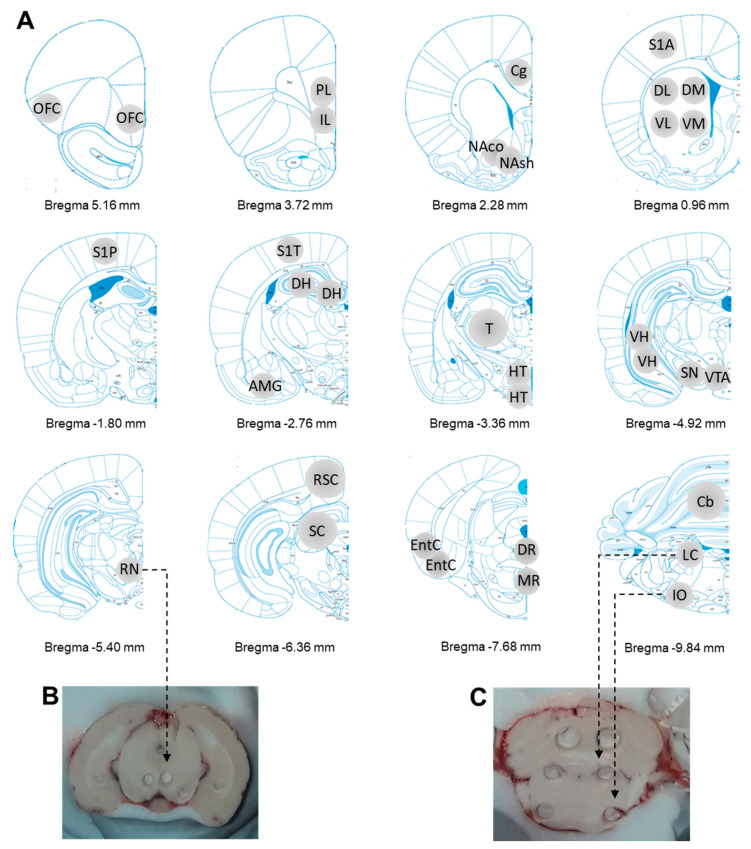
(**A**). Brain regions sampled from Bregma +5.16 to −9.84 mm. OFC = Orbitofrontal cortex, PL = Prelimbic cortex, IL = Infralimbic cortex, Cg = Cingular cortex, NAco = Nucleus accumbens core, Nash = Nucleus accumbens shell, DM = Dorsomedial striatum, DL = Dorsolateral striatum, VL = Ventrolateral striatum, VM = ventromedial striatum, S1A = Primary somatosensory anterior cortex, S1P = Primary somatosensory posterior cortex, S1T = Primary somatosensory trunk cortex, DH = Dorsal hippocampus, AMG = Amygdala, T = Thalamus, HT = Hypothalamus, VH = Ventral hippocampus, SN = Substantia nigra, VTA = Ventral tegmental area, RN = Red nucleus, RSC = Retrosplenial cortex, SC = Superior colliculus, EntC = Entorhinal cortex, DR = Dorsal raphe, MR = Median raphe, Cb = Cerebellum, LC = Locus coeruleus and IO = Inferior olive. OB = Olfactory bulb (not illustrated). (**B**). Example of RN collection in brain. (**C**). Example of LC and IO collection in brain.

**Figure 3 ijms-22-11759-f003:**
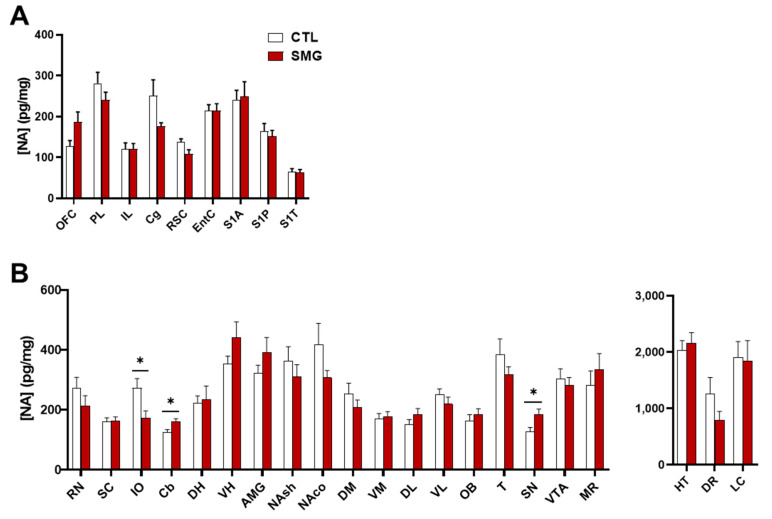
Effect of the SMG on NA tissue content in the brain. NA concentration in pg/mg of tissue for each group is shown in sampled cortical (**A**) and sub-cortical (**B**) brain regions. The values in HT, DR, and LC were separated from the other structures due to the higher levels of NA. All data are presented as mean ± sem (*n* = 12 rats/group). * *p* < 0.05, significant difference between SMG and CTL group (Mann–Whitney test).

**Figure 4 ijms-22-11759-f004:**
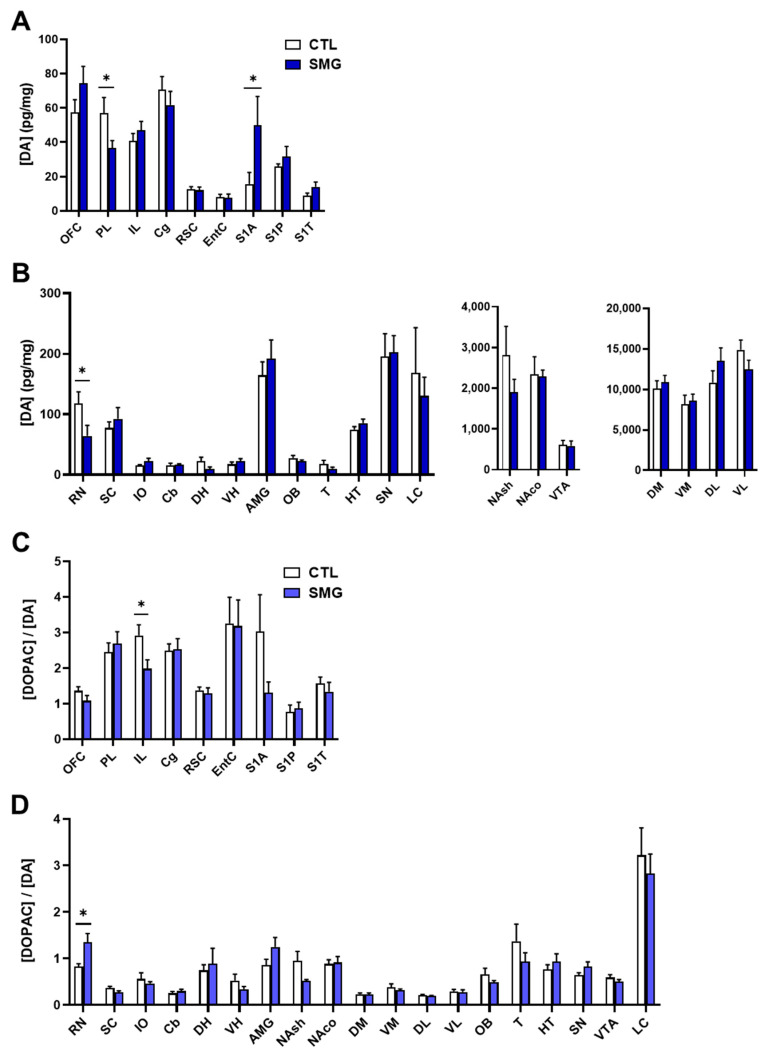
Effect of the SMG on the neurochemistry of the DA system in the brain. DA concentration in pg/mg of tissue for each group is shown in sampled cortical (**A**) and sub-cortical (**B**) brain regions. The ratio of the DOPAC concentration relative to DA concentration is reported in sampled cortical (**C**) and sub-cortical (**D**) brain regions. All data are presented as mean ± sem (*n* = 12 rats/group). * *p* < 0.05, significant difference between SMG and CTL group (Mann–Whitney test).

**Figure 5 ijms-22-11759-f005:**
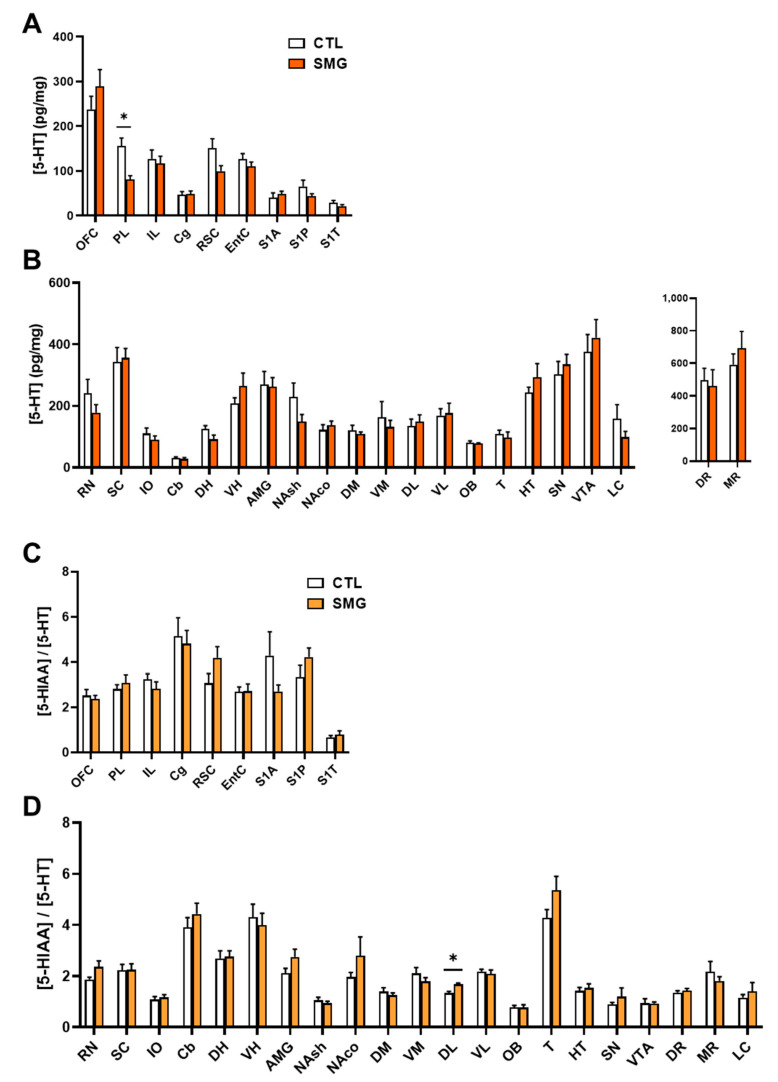
Effect of the SMG on the neurochemistry of the 5-HT system in the brain. 5-HT concentration in pg/mg of tissue for each group is shown in sampled cortical (**A**) and sub-cortical (**B**) brain regions. The ratio of the 5-HIAA concentration relative to 5-HT concentration is reported in sampled cortical (**C**) and sub-cortical (**D**) regions. All data are presented as mean ± sem (*n* = 12 rats/group). * *p* < 0.05, significant difference between SMG and CTL group (Mann–Whitney test).

**Figure 6 ijms-22-11759-f006:**
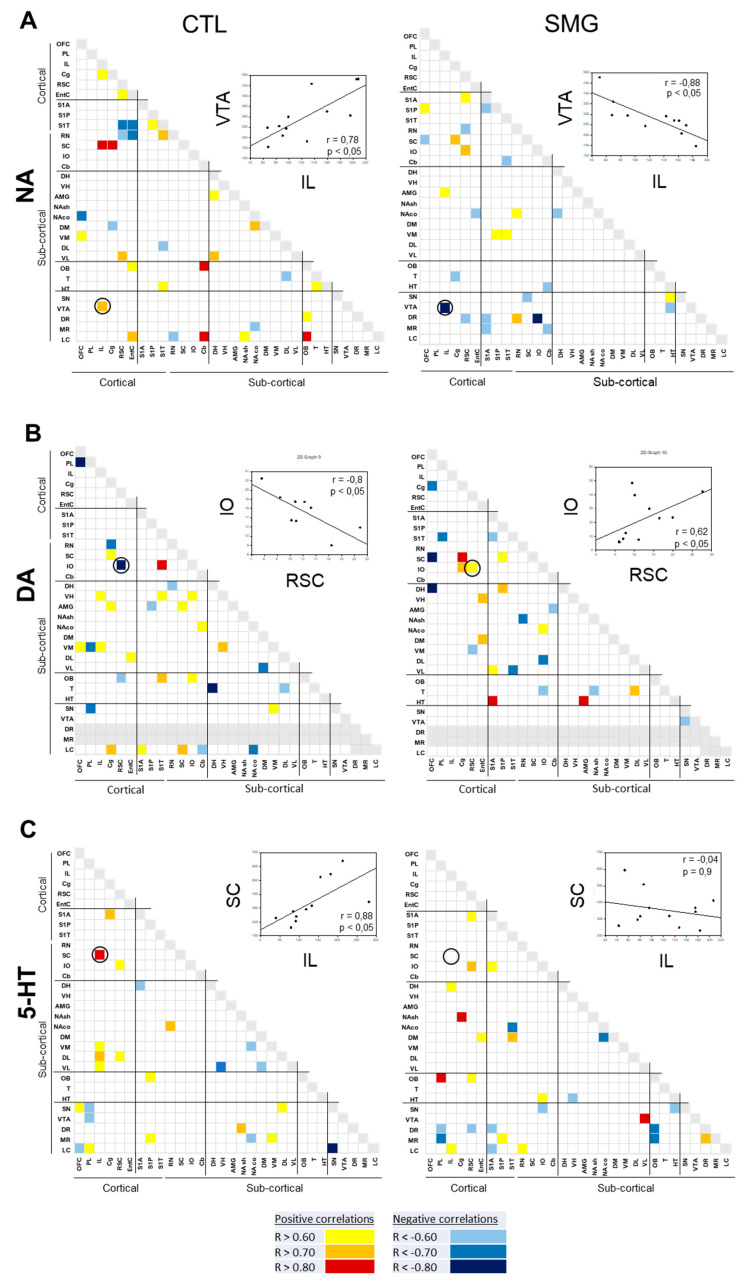
Correlograms for NA (**A**), DA (**B**), and 5-HT (**C**) for the CTL and SMG groups. Positive correlations are reported in yellow (r > 0.6) to red (r > 0.8). Negative correlations are reported in light blue (r < −0.6) to deep blue (r < −0.8). The insert illustrates a representative correlation between brain regions indicated by the circle. Only significant correlations (*p* < 0.05) are reported.

**Figure 7 ijms-22-11759-f007:**
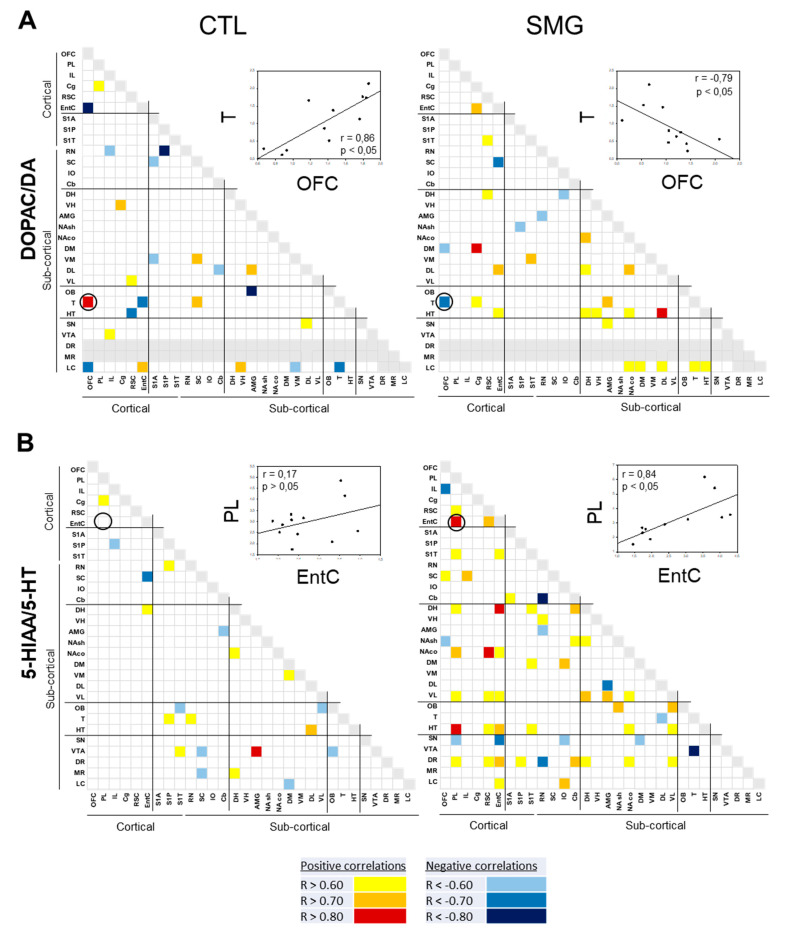
Correlograms for the ratios of DOPAC/DA (**A**) and 5-HIAA/5-HT (**B**) for the CTL and SMG groups. Positive correlations are reported in yellow (r > 0.6) to red (r > 0.8). Negative correlations are reported in light blue (r < −0.6) to deep blue (r < −0.8). The insert illustrates a representative correlation between brain regions indicated by the circle. Only significant correlations (*p* < 0.05) are reported.

**Figure 8 ijms-22-11759-f008:**
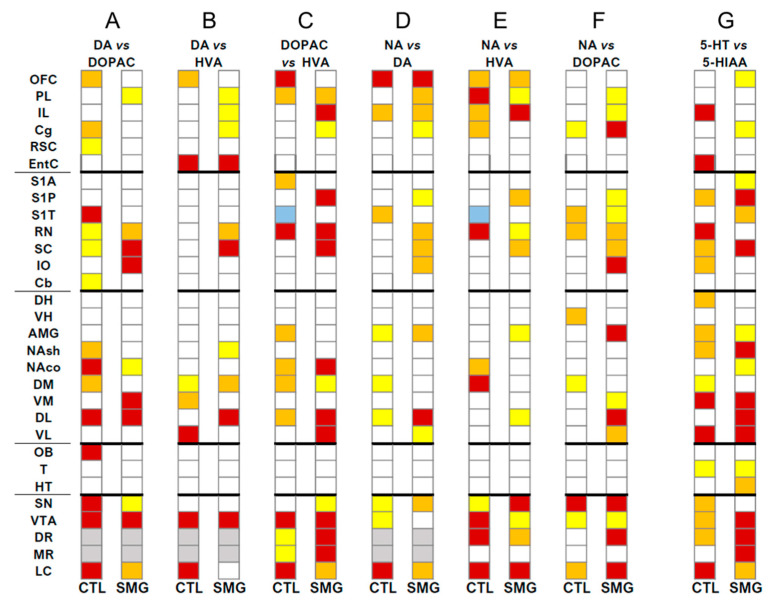
Correlations established between compounds by brain regions for CTL and SMG rats. (**A**). DA vs. DOPAC. (**B**). DA vs. HVA. (**C**). DOPAC vs. HVA. (**D**). NA vs. DA. (**E**). NA vs. HVA. (**F**). NA vs. DOPAC. (**G**). 5-HT vs. 5-HIAA. Positive correlations are reported in yellow (r > 0.6) to red (r > 0.8). Negative correlations are reported in light blue (r < −0.6). Only significant correlations (*p* < 0.05) are reported.

**Figure 9 ijms-22-11759-f009:**
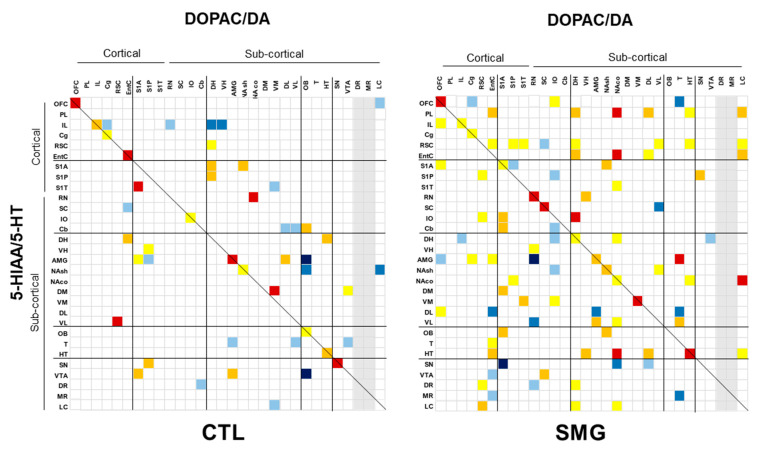
Correlations established between the ratios of DOPAC/DA and 5-HIAA/5-HT across the brain for CTL and SMG rats. Positive correlations are reported in yellow (r > 0.6) to red (r > 0.8). Negative correlations are reported in light blue (r < −0.6) to deep blue (r < −0.8). Only significant correlations (*p* < 0.05) are reported.

## Data Availability

Not applicable.

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
