# Peer review of "Simulated Microgravity Subtlety Changes Monoamine Function across the Rat Brain"

_ijms, 2021, doi:10.3390/ijms222111759_

Round 1

Reviewer 1 Report

This is the revised manuscript (IJMS-1424455) by Alexandra Gros that report the physiological consequences of 7-days microgravity conditions on the distribution of monoamines in functional territories of the Long Evans male rats brain. Authors found that microgravity conditions modified catecholamine tissue contents in vestibular/motor systems, and prelimbic cortex. Authors suggest that change induced by microgravity conditions, although not be pathogenic per se, could likely confer transient vulnerability to develop neuropsychiatric diseases, notably mood disorder.

The article seems well written and and the results are well documented. It contains interesting information about a matter is very important and noteworthy for the research in the topic of astronauts brain adaptive responses during spaceflights, and for astronauts rehabilitation programs. In my opinion the article could be take in consideration for publication after minor revision.

Minor revision:

* Authors explain that microgravity short duration in their study allowed to evaluate the early reorganization of monoaminergic systems. However, it is not clear how these early changes can be maintained (or not) for longer periods, compatible with spaceflights. In Other words, it is possible to assume that neuronal plasticity mechanisms could, in medium-long time period, restore catecholamine’s levels to their initial levels.

In my opinion, this aspect would be important and the authors could better discuss this issue in the discussion sections of the paper

* The authors could better discuss the limits of the study.

* Check the text for grammatical errors and syntax.

Author Response

We thank the reviewer for their comments. The modification in the manuscript are reported in bold blue. Please find enclosed the file with our detailed answer to the comments of the both reviewers.

Briefly, we have modified the discussion section to add the limits of our study and propositions for further experiments to complete this study (lines 467-468 511-512, and 517-520).

Reviewer 2 Report

The manuscript explores the possible effects of the exposure to microgravity for 7 days (short exposure) on the distribution of monoamines in vestibular/motor systems and the possible reorganization of monoamines function across the brain.

The study is interesting and necessary, as it should be taken into account that monoaminergic systems are implicated in numerous neuropsychiatric and neurological diseases. It essential to determine how these systems respond to spaceflights conditions, to microgravity due to the future of those spaceflights. However, the analysis of this system in microgravity conditions is poorly known.

To simulate microgravity, it is used the hindlimb unloading rat model, modified from Morey-Holton method. The hindlimb unloading suspension model has been successfully used in previous works to reproduce the principal effects of microgravity observed at the muscular, bone and cardiovascular levels. The modification by Morey-Holton should be an adaptation for studying the brain, but I am not an expert in these techniques. Authors should briefly explain the rationality of this modification and the suitability of the model. To me, it is strange that there is not change in the levels of corticosterone in the group exposed to microgravity, as the conditions of the model would stress animals.

Methods seem to be correct. Monoaminergic analysis using HPLC, and electrochemical detection is ok, and the sensitivity problems in regions with low concentration are logical. Authors did the best they can, and they describe the difficulties. This is reasonable and acceptable. Statistical data analysis is also correct, although 12 rats are not too much.

Results are abundant, as the study is thorough, precise, and descriptive for 30 brain regions including cortical and subcortical regions. It contains a lot of data, with a remarkable analysis work, and some attempts to solve particular problems related to sensitivity of the detection due to the low content of some monoamines in some of those 30 regions.

The manuscript concluded that microgravity over7 days triggers restricted changes of catecholamines and 5-HT, remodeling the organization of monoaminergic systems in the brain.

Minor points to be addressed:

At line 225, it is stated that the number of correlations across the brain for one monoamine was low. However, 33 correlations were found for NA in CTL, and 23 correlations in the SMG group. Similarly, for DA (34 and 25) and so on. Are those figures considered a low number?. How much correlations should be found to consider a high number?. .

Line 278: Negative correlations are reported in light blue (r < -0.6) to deep blue (r < -0.8). I cannot see any deep blue at the figure 8. Check it, please

The term Pic (I think French) would be replaced by peak. This is in Methods

Author Response

We thank the reviewer for their comments. The modification in the manuscript are reported in bold blue. Please find enclosed the file with our detailed answer to the comments of the both reviewers.

Briefly, we have changed a sentence in the introduction to better explain the expected effects of the microgravity simulation (line 69-70) and add a sentence in discussion section to clarify some difference observed between different studies in the corticosterone levels. Finally, we have modified as you suggested the legend of figure 8 by discarding the allusion to deep blue because deep blue is not present in the figure and “pic” was modified in “peak”
